# Effects of *Cyclocarya paliurus* (Batal.) Extracts on Oxidative Stability and Sensory Quality in Meat Products (Frankfurters)

**DOI:** 10.3390/foods11223721

**Published:** 2022-11-19

**Authors:** Yingying Zhu, Xiaohan Li, Chunyan Da, Panyu Liang, Shuangshuang Jin, Changbo Tang

**Affiliations:** 1Key Laboratory of Meat Processing and Quality Control, MOE, Key Laboratory of Meat Processing, MOA, Jiangsu Collaborative Innovation Center of Meat Production, Processing and Quality Control, College of Food Science and Technology, Nanjing Agricultural University, Nanjing 210000, China; 2Engineering Research Center of Magnetic Resonance Analysis Technology, Department of Food Nutrition and Test, Suzhou Vocational University, Suzhou 215104, China

**Keywords:** *Cyclocarya paliurus*, flavonoid, meat product, lipid oxidation, sensory quality, frankfurter

## Abstract

Oxidation is one of the most common causes of the deterioration of meat and meat products. At the same time, synthetic antioxidants are becoming less accepted by consumers due to the potential health hazards they might cause. Therefore, a new trend to substitute these synthetic antioxidants with natural antioxidants has emerged. This study adds flavonoid extracts from *Cyclocarya paliurus* (*C. paliurus*) as a natural antioxidant for meat products (Frankfurters). The results showed that flavonoid extracts from *C. paliurus* had strong antioxidant and antibacterial activity. This is proportional to concentration, and the addition of extracts could significantly (*p* < 0.05) delay the lipid oxidation in the samples. In addition, we did not observe hazardous effects on the samples’ pH and texture as a result of adding flavonoid extracts. We observed that flavonoid extracts from *C. paliurus* at concentrations of 0.06% and 0.12% did not affect the color and sensory evaluation of the samples. At a concentration of 0.18% and 0.24%, the flavonoid extracts had a negative impact on the color and sensory evaluation of the samples, likely due to the yellow-brown color of the extract itself. The findings showed that a low concentration of 0.12% flavonoid extracts from *C. paliurus* in meat products could effectively prevent lipid oxidation without affecting the sensory quality.

## 1. Introduction

Lipid oxidation has long been recognized as the leading cause of undesirable effects on meat and meat products’ quality, acceptability, and shelf-life. Lipid oxidation results in oxidative off-flavor, discoloration, and spoilage of meat and meat products [1,2]. The oxidation process involves the degradation of PUFAs (Polyunsaturated fatty acids) and the generation of free radicals, which can lead to the production of rancid odors and changes in the color and texture of meat [3]. Therefore, interest in preventing and delaying lipid oxidation in meat products by using antioxidant agents from both natural sources (plants and fruits) and synthetic (i.e., BHT (butylated hydroxytoluene), BHA (butylated hydroxyanisole), TBHQ (tertiary butylhydroquinone), etc.) has increased. Subsequently, it improves the shelf life and nutritional value of meat and meat products [4,5,6]. Certain available synthetic antioxidants have been suspected to be toxic and have hazardous effects on human health; consequently, their use in food has been restricted [7]. Additional disadvantages to synthetic antioxidants are that they can cause discoloring and off-flavors of products. With the growing interest in purchasing natural products from consumers, how to substitute synthetic antioxidants with natural antioxidants has received the most attention from researchers and meat processors [8,9,10].

Natural antioxidants are mainly derived from spices, plants, fruits, and vegetable skin residue extracts. Numerous studies have evaluated natural substances as antioxidant additives in meat products and proved their effectiveness. Estevez et al. found that adding rosemary essential oil to frankfurters delayed oxidation problems and reduced tenderness during refrigeration [11]. Tran et al. added guava leaf extract to fresh pork sausage and found that it could effectively delay lipid oxidation at 4000, 5000, and 6000 ppm [12]. Rehana et al. found that incorporating microencapsulated Rosemary or mint oil could prevent significant changes in PUFAs and reduce the formation of cholesterol oxidation products (COPs) in meat emulsions [13]. Adding berry (e.g., blackcurrant, raspberry, or chokeberry) extracts to meat products has improved their sensory attributes while increasing their nutritional value and health safety [14]. Natural antioxidants can prevent significant changes in PUFAs, improve the nutritional value of food and prolong shelf-life.

*Cyclocarya paliurus* (Batal.) Iljinskaja (*C. paliurus*), commonly known as the “sweet tea tree”, is a well-known edible and medicinal plant cultivated in the misty highlands of southern China. In recent years, *C. paliurus* has gained increasing interest due to its wide range of biological activities and antioxidant effects, such as antihypertensive activity, hypolipidemic, hypoglycemic activity, enhancement of mental efficiency, and antioxidant activity [15,16,17]. These activities have been attributed to the main active ingredients in *C. paliurus* leaves, such as flavonoids, polysaccharides, triterpenoids, steroids, saponins, and phenolic acid compounds [16,18]. Amongst these compounds, flavonoids were one of the main active compounds in the *C. paliurus* leaves. Xie et al. found that the DPPH radical scavenging ability of flavonoids (at the concentration of 0.1–0.8 mg/mL) from *C. paliurus* was always better than that of butylated hydroxytoluene (BHT) [19]. Regarding antibacterial properties, 80 μg/mL flavonoids from *C. paliurus* had inhibited effects on Staphylococcus aureus, Salmonella, and Escherichia coli [20]. Furthermore, the phenolic content of leaves is usually higher than that of fruits [21]. 

Consumers expect meat products to be nutritious, safe, convenient, and of good sensory quality. In the meat industry, there is growing interest in using innovative processing methods to reformulate products and replace synthetic additives with natural bioactive compounds to minimize health concerns and improve the overall organoleptic, nutritional, and health properties of processed meats. Frankfurters are an emulsion-type meat product containing 20~30% of fat, unfermented, and consumed worldwide. They have a short shelf life as lipids are vulnerable to oxidative damage from reactive oxygen species (ROS), particularly when exposed to light [22,23]. The attack of ROS on muscle leads to lipid and protein oxidation and thus affects color, texture, and sensory changes. Many studies have focused on adding natural extracts to frankfurters as an alternative to synthetic antioxidants. The extracts are from, and a positive effect has been reported on the TBARS value of meat products during storage [14,24].

The flavonoid extracts from *C. paliurus* have specific antioxidant and antibacterial properties that could be used in the meat processing industry as a natural function antioxidant. This study aimed to assess the possibility of adding flavonoid extracts from *C. paliurus* at different concentrations (T1: 0.06%, T2: 0.12%, T3: 0.18%, T4: 0.24%) as a natural antioxidant to improve the physicochemical and sensory properties of cooked meat product (Frankfurters) over a refrigerated storage period of 21 days. 

## 2. Materials and Methods

### 2.1. Material Preparation 

Flavonoid extracts from *C. paliurus*, a brown-yellow fine powder soluble in water has the unique flavor of *C. paliurus*, which were purchased from Lanzhou Waters Biotechnology Co., Ltd. (Lanzhou, China); Sodium D-isoascorbate was purchased from Shiyao Group Weisheng Pharmaceutical (Shijiazhuang) Co., Ltd. (Shijiazhuang, China); Pig hind leg and pig back fat was purchased from Jiangsu Nanjing Yurun Group. White granulated sugar, salt, white pepper powder, and nutmeg powder were all food-grade and commercially available, sodium tripolyphosphate was purchased from Shanghai Taixin Industry Co., Ltd. (Shanghai, China), and collagen casing (22 mm) was purchased from Liu Zhou Honsen Collagen Casing Co., Ltd. (Liuzhou, China). Anhydrous ethanol and methanol were purchased from Sinopharm Chemical Reagent Co., Ltd. (Shanghai, China). 

### 2.2. DPPH Scavenging Activity of the Extract

The DPPH free radical scavenging rate was determined by following the instructions of the DPPH free radical scavenging capacity kit (Nanjing Jiancheng Bioengineering Research Institute). First, we take one working solution powder, add 40 mL of pure ethanol, shake it well, and store it at 4 °C away from the light. Then, we weighed the precise amount of *C. paliurus* flavonoid extracts and prepared liquid samples with the concentration of 0.6 mg/mL, 1.2 mg/mL, 1.8 mg/mL, and 2.4 mg/mL with ultrapure water, and 0.025% sodium D-isoascorbate was prepared. Following that, we take two 1 mL centrifuge tubes for each group, add 400 μL of liquid sample, 600 μL of 80% methanol for one as the control group, and 600 μL of working solution for the other as the test group. We also add 400 μL, 80% methanol, and 600 μL working solution into the blank tube as the blank control. All treatment groups were covered to avoid light for 30 min, then 200 μL were absorbed into 96 healthy enzyme standard plates, and the absorbance was measured at 517 nm. The DPPH radical scavenging rate (%) was calculated using the following Equation (1): (1)$$\mathrm{DPPH}\;\mathrm{radical}\;\mathrm{scavenging}\;\mathrm{rate}\left(\%\right)=\left(1-\frac{A_{1}-A_{2}}{A_{0}}\right)\times 100\%$$
where *A*_1_ is the absorbance of the test group, *A*_2_ is the absorbance of the control group, and *A*_0_ is the absorbance of the blank group. All measurements were performed in triplicate.

### 2.3. Preparation of Cooked Meat Products (Frankfurters)

To prepare the cooked meat product (frankfurters), pork hindquarter and pig back fat were purchased, the fascia was removed, separately ground in a meat grinder, and organized into six treatment groups (C, VC, T1, T2, T3, and T4). The composition of frankfurters is listed in Table 1, all ingredients have been mixed, and the antioxidants (sodium D-isoascorbate and flavonoid extracts from *C. paliurus*) have been added. All the elements were chopped with a chopping machine (K15E, TALSA, Liège Belgium, Spain) and poured into 22 mm collagen casings with a sausage machine (H15PA, TALSA, Liège Belgium, Spain), tied at every 10 cm, and steamed in a fuming and boiling machine to maturity. The steam frankfurters were cooled to below 15 °C, then vacuum packed in a vacuum packaging machine (DC-800, Sealed Air, North Carolina, United States), sterilized, and stored in a 4 °C refrigerated warehouse. Group C contained no antioxidants as a negative control. Group VC contained 0.025% sodium D-isoascorbate as a positive control. Groups T1, T2, T3, and T4 contained 0.06%, 0.12%, 0.18%, and 0.24% flavonoid extracts from *C. paliurus* as different treatments, respectively. All samples were evaluated on the 1st, 7th, 14th, and 21st days.

### 2.4. Determination of Malondialdehyde (MDA) Content

Thiobarbituric acid reactive substances are one of the most intuitive indicators of lipid oxidation. The oxidative stability of the frankfurters was based on measurements of the malondialdehyde (MDA) concentration by a malondialdehyde detection kit (Suzhou Comin Biotechnol Technology Co., Ltd., Suzhou, China). The content of MDA was determined following the steps of the detection kit manual: ① weigh 0.2 g of meat sample into a 5 mL centrifuge tube, add 2 mL of extract, and homogenized in an ice water bath; ② the homogenate was centrifuged at 8000 *g* for 10 min at 4 °C with a centrifuge (D3024R, Dlab Scientific Co., Ltd., Beijing, China), the supernatant was taken and placed on ice until tested; ③ take a 1.5 mL centrifuge tube, accurately absorb 0.3 mL reagent 1 and 0.1 mL supernatant into the test tube, mix well, and water bath at 95 °C for 30 min; ④ placed the mixed liquid in the ice water for cooling down, and centrifuge at 10,000 *g* for 10 min; ⑤ 200 μL of the supernatant was put into a 96-well plate, and the absorbance was measured at 532 nm and 600 nm. An automatic biochemical analyzer detected the total protein concentration (BK-280, BIOBASE, Jinan, China). Take an appropriate amount of supernatant and use the automatic biochemical analyzer to detect the total protein concentration of the sample (BK-280, BIOBASE, Jinan, China). *MDA* content was calculated using the following Equation (2): (2)MDA content (nmol/mg prot)=51.6×ΔA÷Cpr
where ∆*A* = A532-A600 and *Cpr* is the sample protein concentration (mg protein/mL). All measurements were performed in triplicate.

### 2.5. Microbial Experiment

The total viable counts (TVC) were determined per the national food safety standard GB4789.2-2016 using the aerobic plate counting method [25]. Plates were incubated at 37 °C for 48 h of frankfurters samples with different antioxidants on days 0, 7, 14, and 21. The result was expressed as log_10_ CFU·g^−1^. *Escherichia coli* was determined following the national food safety standard GB4789.2-2016 [26]. *Escherichia coli* was inoculated on the respective nutrient medium and incubated at 37 °C for 24 h of frankfurters samples with different antioxidants on days 0, 7, 14, and 21. All measurements were performed in triplicate.

### 2.6. pH

A portable digital pH meter (Testo 205-PH, Testo, Lenzkirch, Germany) was used to detect the pH of the frankfurters. Before measurements, a standard buffer solution (pH = 4.01, pH = 7.00) was used for a two-point calibration. The pH value was measured on the 1st, 7th, 14th, and 21st days. All measurements were performed in triplicate.

### 2.7. Color

The color difference is measured by a portable color difference meter with an 8 mm aperture and D65 light source (CR-400, Konica Minolta, Tokyo, Japan). The colorimeter was calibrated with a standard whiteboard (Y = 87.9, x = 0.3133, y = 0.3196) before use, and the frankfurter was cut into columns 1 cm high and 22 mm in diameter. Each frankfurter was measured twice at the front and rear, and the values for *L**, *a**, and *b** were recorded. The color difference was measured on the 1st, 7th, 14th, and 21st days. The previous study has indicated that a total color difference (ΔE) of approximately 1 is discriminable by consumers [27]. The total color difference ΔE was calculated using the following Formula (3): (3)$$\Delta E=\sqrt{\Delta L^{*2}+\Delta a^{*2}+\Delta b^{*2}}$$
where Δ*L**, Δ*a**, Δ*b** were calculated by the difference between stored samples and 1st day samples. All measurements were performed in triplicate.

### 2.8. Texture

The determination of texture was referred to the method of Zhou et al. for appropriate modification [28]. We took frankfurters sample from six different testing groups, peeled off the casings, and cut 2 cm high and 22 mm diameter cylinders, to conduct two chewing tests with a texture analyzer (TA.XT Plus texture analyzer, Stable Micro Systems, UK) and a probe of model P/50 with the diameter of 50 mm. The speed before the test was 2 mm/s, the speed during the test was 2 mm/s, the speed after the test was 5 mm/s, the trigger force was set as 5 g, and the compression ratio was 35%. The firmness, springiness, and chewiness were recorded, and the texture was determined on the 1st, 7th, 14th, and 21st days. All measurements were performed in triplicate.

### 2.9. Sensory Evaluation

The method of sensory reference method has been amended appropriately for our previous experiment [29]. A panel of ten trained tasters evaluated the sensory properties of frankfurter samples, including color, flavor, juiciness, taste, and overall acceptability. Referring to Djekic et al. [30], the members of the sensory evaluation group had a great influence on the results. Training and performance prescreening of the sensory panel are very important for the result of the food sensory evaluation. Panelists were chosen based on previous experience in evaluating frankfurters and trained in accordance with ISO 11132: 2021 prior to evaluating in order to reduce the differences caused by individuals. The frankfurters were heated in a microwave oven for 1 min, cut into 5 mm thin slices, and numbered randomly. Each member tasted and scored the color, flavor, juiciness, taste, and overall acceptability of the frankfurters in the evaluation form using a score between 1 to 9, and all the data were collected afterward. Sensory evaluation was carried out on the 1st, 7th, 14th, and 21st days.

### 2.10. Statistical Analysis

SAS 9.4 software was used for statistical analysis of the collected data, and unidirectional ANOVA and multiple Duncan comparisons were used to test the significance of the differences. The results were presented as mean ± standard deviation.

## 3. Results and Discussion

### 3.1. DPPH Scavenging Activity of Extracts

To investigate the effects and influence of different antioxidants on meat products, the DPPH scavenging activity of different antioxidants has been determined. The DPPH free radical scavenging rate can effectively show the total antioxidant activity in natural products [31]. Flavonoids from *C. paliurus* are a natural extract that has been proven to have certain antioxidant properties [32]. To determine the relationship between the concentration of flavonoids and their antioxidant effect, the extracts were diluted into 0.06%, 0.12%, 0.18%, and 0.24% solutions, and DPPH free radical scavenging assay was performed. The results are shown in Table 2. The higher the concentration of flavonoid extracts from *C. paliurus*, the more efficient the DPPH scavenging activity is. The DPPH scavenging ability of the extract was significantly increased (*p* < 0.05) and enhanced rapidly when the concentration increased from 0.06% to 0.12%, the growth rate of DPPH scavenging ability was slowed down when the concentration from 0.12% to 0.24% and with no difference between T3 and T4 (*p* > 0.05). Sodium D-isoascorbate has an excellent antioxidant effect and is widely used in meat products [33]. DPPH has a scavenging rate of 0.025% sodium D-isoascorbate was 63.11 ± 0.49%, significantly higher than T1 but lower than T2. The result showed that the concentration exceeding 0.12% of flavonoid extracts from *C. paliurus* was more effective at DPPH scavenging activity than 0.025% sodium D-isoascorbate.

### 3.2. Color Analysis

Color is one of the most important meat quality attributes when consumers are concerned [34]. Δ*E* could be reflected in the changes in the overall color of frankfurters during storage. *L** exhibits the degree of lightness and represents the brightness intensity. *a** exhibits the degree of redness and represents the redness intensity, and *b** exhibits the degree of yellowness and represents the yellowness intensity. Table 3 describes the changes of *L**, *a**, and *b** of frankfurters on the 1st, 7th, 14th, and 21st days, and ΔE was calculated on the 7th, 14th, and 21st days of storage in various test groups. Over the storage period, Δ*E* was increased and then declined apart from the C group. On the 7th day, Δ*E* was more than 1 in groups C and T4 but lower than 1 in groups VC, T1, T2, and T3. The result showed that adding antioxidants delayed the color change of frankfurters in the early storage period. On the 14th day, ΔE was more than 1 in six groups, groups C and T4 with the lowest ΔE, followed by groups T2, VC, T3, and T1, which may be caused by fat oxidation and a dark brown color of extract. On the 21st day, ΔE showed a downward trend in five treatment groups. Over the storage period, *L** increased initially and then declined apart from the T2 group. Compared to group T1, groups T2, T3, and T4 maintained low *L**, possibly due to the enhanced water-holding capacity (WHC) of frankfurters, which reduces light reflection [35]. Therefore, adding higher flavonoid extracts may improve WHC in frankfurters. In comparison, adding a lower concentration of flavonoid extracts has no significant effect on the WHC in frankfurters. The hemoprotein, myoglobin, and hemoglobin in frankfurters will undergo an oxidation reaction, resulting in a color shift from red to brown [36]. In addition, the extract contains phenolic substances, and the higher concentration of the extract, the higher composition of phenolic substances [19]. Phenolic substances could be oxidized into quinones during storage, thus reducing the *L** of frankfurters. Meanwhile, fat oxidation also adversely affects meat color. Throughout the storage period, group VC consistently maintained the highest *a** value, significantly different (*p* < 0.05) from other test groups except on day 1. The result indicated that adding sodium D-isoascorbate could better maintain the color stability of meat products than other test groups. Among groups T1, T2, T3, and T4, *a** initially increased, then declined as the concentration of flavonoid extracts increased; group T2 has the highest *a** value, indicating that flavone extract may have a significate contribution in color protection; however, this color protection did not show a trend of increasing with the concentration of flavonoid extracts. Except for group T4, the higher the extract concentration, the greater the *b** value. The *b** value for all treatment groups increased throughout the storage period. On the 1st day, b* of frankfurters showed a trend of increasing with the concentration of flavonoid extracts, due to the influence of extracts’ color. On the 14th day, the *b** value of group T1 was lower than group C but gradually increased and eventually exceeded group C with the added extract concentration. Compared to other groups, excluding group VC, *b** in group T1 was the lowest, possibly due to the higher antioxidant effect of the extracts. Flavonoid extracts inhibited fat and protein oxidation, maintaining the color stability of frankfurters [36]. Manzoor et al. found that mango peel extract was added to chicken sausage, and *b** gradually decreased during the whole storage period [37]. Nevertheless, TBARS values were significantly lower, excluding the yellow color as a result of lipid oxidation [38]. The lower *a** and higher *b** in the test groups may be due to the yellow-brown color of the flavone extract affecting the color of the frankfurters.

### 3.3. pH Analysis

Table 4 shows the changes in pH values over the storage period in different test groups. All groups exhibited minor changes in pH over the storage time (*p* > 0.05). The pH of the group VC samples was always the highest over the storage period, which was significantly higher (*p* < 0.05) than the other five test groups. On the first day, all treatment groups showed a higher pH than group C except group T4, which may be related to a higher concentration of polyphenols contained in the extract, which may be decreased the pH of frankfurters. Meanwhile, microbial fermentation will also reduce the pH due to the presence of microorganisms in group T4. Compared with the 1st day, the pH of the samples on the 7th day declined; the pH of the group VC samples was the highest, with an average value of 6.34. In addition, the pH of group C was significantly different from those samples from groups tested with flavone extract, except for group T1 (*p* < 0.05). On the 14th day, the pH of groups T1 and T4 samples was lower, averaging 6.25 and 6.24, respectively. On the 21st day, all treatment groups showed little change in pH compared with the 14th day except group T4. In addition, as storage time increased, the pH value of each test group gradually decreased and then increased except for group T4, which may be caused by the higher antioxidant effect of the extract led to higher oxidation stability of frankfurter samples and reducing the oxidation [9,39]. All treatment groups (except group T4) showed little change in pH during the storage time, indicating that the addition of flavonoid extracts from *C. paliurus* had no significant (*p* > 0.05) effect on pH. Rapid reprocessing micro-organisms can increase pH during storage in the samples at the later stage field [40].

### 3.4. Textural Analysis

Textural analysis was performed throughout the storage period to determine whether the addition of flavonoid extracts affected the frankfurters’ final firmness, springiness, and chewiness. The result is presented in Table 5. There was no significant difference in firmness, springiness, and chewiness between all groups on the 1st day. The data fluctuated very little between the 1st day to the 21st day (*p* > 0.05), indicating that antioxidant supplementation had no significant impact on springiness or chewiness except for samples in group T4. There were significant differences in firmness between each test group after 7 days of storage, and group C always had the highest firmness during storage time. The previous study showed that phenolic compounds extracted from plants could reduce the firmness of cooked meat products during cold storage. This allows the meat to obtain better emulsion stability through the antioxidant effects of the plant extracts on the lipids and proteins [41]. Furthermore, oxidation leads to the separation of fat and water, increasing the firmness of cooked meat products [11,41]. Our results indicate that adding antioxidants helps improve the tenderness of frankfurters, and flavone extract positively affects the texture properties of frankfurters during storage time.

### 3.5. MDA Analysis

MDA analysis was performed throughout the whole storage to determine whether adding flavonoid extracts impacts lipid oxidation during the storage period. The result is presented in Figure 1. The MDA content of the samples showed an increase initially and then declined in all groups over the storage period. On the 1st day of storage, group VC had the highest content of MDA compared to other treatment groups, with an average value of 0.55 nmol/mgprot, which was slightly higher than group C. and significantly higher than that of the rest four test groups with flavonoid extracts T1, T2, T3 and T4 (*p* < 0.05). On the 7th day of storage, MDA content in group C was drastically increased and considerably higher than the other four test groups with different concentrations of *C. paliurus* extracts (*p* < 0.05). Compared to other treatment groups, groups T3 and T4 always have the lower MDA content, which showed that the higher concentration of flavonoid extracts, the more effective the antioxidant activity of meat products. On the other hand, MDA content also increased in all the test groups with different concentrations of flavonoid extracts during the process, which is consistent with the results from the Zhang et al. study [36]. This can be caused by steroid ketones, carbohydrates, amino acids, amides, pyrimidines, and vitamins produced rapidly in the sample during heating. These substances can react with TBA and then increase the content of MDA. Results are consistent with the research of Rahimeh et al. [39]. Our results indicate the presence of lipid oxidation in samples during storage; adding the natural extracts can effectively delay lipid oxidation in the frankfurters, demonstrating inhibition of the lipid oxidation effect with the addition of flavonoid extracts.

### 3.6. Microbial Examination

The microbiological test was performed on all samples on the 1st, 7th, 14th, and 21st days. The result is presented in Table 6. *Escherichia coli (E. coli.)* was not observed in the analyzed samples during the entire testing period, indicating no secondary contamination of *E. coli* during the experiment. The total number of colonies (TVC value) increased over time while in storage in all treatment groups. Some colonies were detected in groups T3 and T4 on the first day may be caused by incomplete sterilization. On the 7th and 14th day, no colonies were detected in group VC; group C had the highest mean values for TVC, the values of TVC in groups T3 and T4 were significantly higher than in groups T1 and T2 (*p* < 0.05), might be caused by the initial colonies. In contrast, although the total number of initial colonies in groups T3 and T4 were higher, the trend of TVC values increase was slower than in group T1 and T2 in the storage period, suggesting that the higher concentration of extracts, the better antimicrobial ability. The growth rate of TVC values in the treatment groups with the addition of extracts and antioxidants was slower than that in group C, indicating that the antioxidant and flavonoid extract from *C. paliurus* had specific antibacterial activity [42]. It can be concluded that the flavonoid extracts from *C. paliurus* can be used as natural antibacterial agents in meat products.

### 3.7. Sensory Evaluation

The sensory quality of the product is one of the most important considerations for consumers. Adding additives may change the flavor, color, and taste of products, affecting customer satisfaction. Frankfurters were evaluated on days 1, 7, 14, and 21 of storage (Table 7). The objective was to observe the sensory changes of frankfurters during storage. There were no noteworthy changes in sensory between days 1 and 7 on all test groups. Group VC had a lower score on the overall acceptability, likely due to the ability of sodium D-isoascorbate to bind water and create bubbles in the meat, affecting the batter of the product. Groups T1 and T2 had a higher overall acceptability score, with group T2 being the highest. After 14 days, we observed that the scores for all sensory indicators dropped in all groups, with the highest score in group T2, followed by T1, T3, and T4. Group VC group scored the highest in color and flavor, and the addition of sodium D-isoascorbate had a good maintenance effect on the color and flavor of the frankfurters. Jaberi et al. added barberry extracts into chicken frankfurters at concentrations of 0.75, 1.5, and 3% and found a significant difference in flavor between the test groups [39]. Stobnicka et al. used swamp cranberry fruit and pomace extract in pork burgers and found that adding two extracts with a concentration of 2.5% in pork burgers did not affect the sensory quality [2]. Overall, natural extracts can even improve the sensory quality of meat products to a certain extent [2,43]. Our study demonstrated that adding flavonoid extracts has improved the sensory quality of frankfurters.

## 4. Conclusions

Our study observed how different concentrations of flavonoid extracts from *C. paliurus* affect the overall quality, fat oxidation, TVC, and sensory of vacuum-packed cooked meat products (Frankfurters) over 21 days of storage. The DPPH scavenging activity showed that flavonoid extracts were an effective antioxidant, and the greater the concentration, the greater the antioxidant capacity. Compared to the negative control group C, the four different extract concentrations can significantly reduce the MDA concentration in samples (*p* < 0.05). The antioxidant effect of the low-concentration extract (0.06% and 0.12%) is equivalent to sodium D-isoascorbate. In comparison, the antioxidant effect of the high-concentration extract (0.18% and 0.24%) is significantly higher than that of sodium D-isoascorbate, indicating that the antioxidant capacity of flavonoid extracts is related to the concentration. The flavonoid extracts from *C. paliurus* had a certain antibacterial effect, and the higher concentration of extracts, the better the antimicrobial ability. Adding flavonoid extracts help to improve the tenderness of frankfurters during storage. When extracts were added, no harmful effects were observed on the quality and pH value. In the sensory evaluation towards the storage period, samples with low extract concentration had better color and appearance parameters scores. Samples with flavonoid extracts at concentrations of 0.18% and 0.24% had lower color and sensory evaluation scores, which might be caused by the yellow-brown color of the extract itself. The findings conclude that the low concentration (0.12%) of flavonoid extracts from *C. paliurus* in meat products could effectively inhibit lipid oxidation without affecting sensory quality.

## Figures and Tables

**Figure 1 foods-11-03721-f001:**
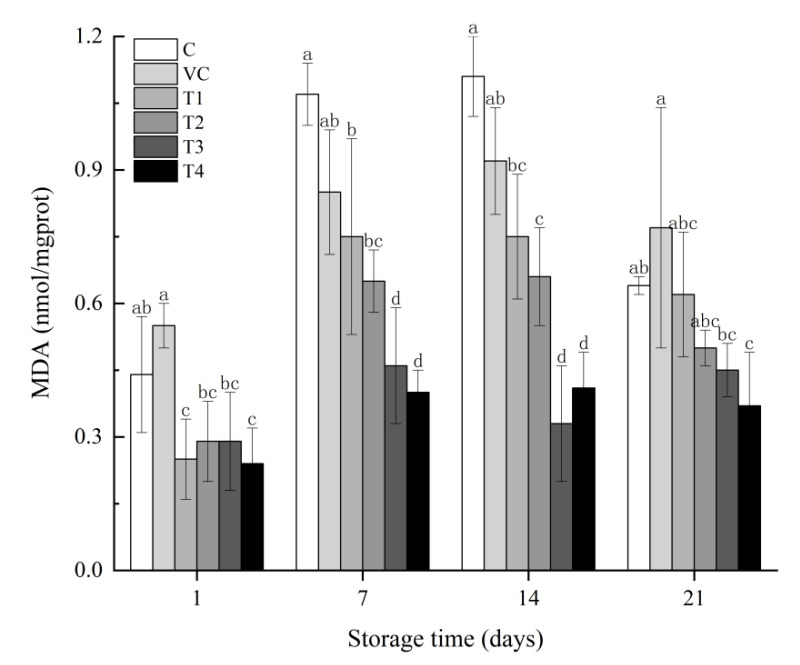
Effect of adding antioxidants on TBARS values of frankfurters during storage at 4 °C (n = 6). Note: Standard error bars are indicated. Different letters on the bars represent significant differences (*p* < 0.05) in samples. Treatments: C: control, without antioxidant; VC: 0.025% sodium D-isoascorbate; T1 T2, T3, and T4: contained 0.06%, 0.12%, 0.18%, and 0.24% flavonoid extracts from *C. paliurus*, respectively.

**Table 1 foods-11-03721-t001:** Ingredients of raw-cooked meat products (frankfurters).

Ingredient	C	VC	T1	T2	T3	T4
Lean meat (g)	1500	1500	1500	1500	1500	1500
Fat meat (g)	600	600	600	600	600	600
ice water (g)	500	500	500	500	500	500
salt (g)	26	26	26	26	26	26
White sugar (g)	3.75	3.75	3.75	3.75	3.75	3.75
Sodium tripolyphosphate (g)	7.5	7.5	7.5	7.5	7.5	7.5
White pepper powder (g)	7.2	7.2	7.2	7.2	7.2	7.2
Nutmeg powder (g)	2.4	2.4	2.4	2.4	2.4	2.4
Sodium D-isoascorbate (g)	-	0.65	-	-	--	-
*C. paliurus* extract (g)	-	-	1.56	3.12	4.68	6.24

Note: “-” in the table represent the absence of individual ingredient for each treatment. Treatments: C: control, without antioxidant; VC: 0.025% sodium D-isoascorbate; T1 T2, T3, and T4: contained 0.06%, 0.12%, 0.18%, and 0.24% flavonoid extracts from *C. paliurus*, respectively.

**Table 2 foods-11-03721-t002:** DPPH scavenging activity of VC and extracts (n = 3).

	VC	T1	T2	T3	T4
DPPH scavenging rate (%)	63.11 ± 0.49% ^c^	51.25 ± 0.82% ^d^	80.32 ± 1.95% ^b^	87.60 ± 1.37% ^a^	90.79 ± 0.34% ^a^

Note: Results were expressed as mean ± standard deviation, different superscript letters represent significant differences (*p* < 0.05). Treatments: VC: 0.025% sodium D-isoascorbate; T1 T2, T3, and T4: contained 0.06%, 0.12%, 0.18%, and 0.24% flavonoid extracts from *C. paliurus*, respectively.

**Table 3 foods-11-03721-t003:** Changes of color parameters and total color difference ΔE of frankfurters during storage at 4 °C (n = 6).

	Group	Day 1	Day 7	Day 14	Day 21
*L**	C	73.01 ± 0.32 ^b^	74.05 ± 0.61 ^b^	71.96 ± 0.22 ^b^	71.96 ± 0.06 ^b^
lightness	VC	72.90 ± 0.16 ^bc^	73.13 ± 0.05 ^c^	71.25 ± 0.21 ^c^	71.71 ± 0.21 ^b^
	T1	74.88 ± 0.32 ^a^	75.01 ± 0.03 ^a^	72.88 ± 0.45 ^a^	73.53 ± 0.14 ^a^
	T2	72.20 ± 0.62 ^cd^	71.75 ± 0.12 ^e^	70.51 ± 0.13 ^d^	70.99 ± 0.17 ^c^
	T3	72.16 ± 0.53 ^cd^	72.28 ± 0.32 ^d^	70.45 ± 0.29 ^d^	71.05 ± 0.19 ^c^
	T4	71.78 ± 0.42 ^d^	72.85 ± 0.22 ^c^	70.74 ± 0.24 ^d^	70.86 ± 0.78 ^c^
*a**	C	4.78 ± 0.06 ^bc^	5.16 ± 0.12 ^c^	5.41 ± 0.27 ^b^	5.95 ± 0.26 ^b^
redness	VC	5.57 ± 0.08 ^a^	6.35 ± 0.15 ^a^	6.40 ± 0.11 ^a^	6.79 ± 0.05 ^a^
	T1	4.33 ± 0.13 ^d^	4.60 ± 0.20 ^e^	4.84 ± 0.42 ^c^	4.85 ± 0.17 ^c^
	T2	5.52 ± 0.09 ^a^	5.55 ± 0.05 ^b^	5.73 ± 0.44 ^b^	5.82 ± 0.35 ^b^
	T3	4.88 ± 0.07 ^b^	4.90 ± 0.03 ^d^	4.69 ± 0.04 ^c^	4.46 ± 0.07 ^d^
	T4	4.62 ± 0.13 ^c^	4.60 ± 0.10 ^e^	4.49 ± 0.08 ^c^	4.35 ± 0.11 ^d^
*b**	C	8.59 ± 0.01 ^b^	8.58 ± 0.04 ^b^	9.51 ± 0.07 ^d^	8.92 ± 0.19 ^c^
yellowness	VC	7.73 ± 0.18 ^c^	8.14 ± 0.16 ^c^	8.82 ± 0.11 ^e^	8.35 ± 0.23 ^d^
	T1	7.75 ± 0.09 ^c^	8.50 ± 0.17 ^b^	9.39 ± 0.21 ^d^	9.15 ± 0.08 ^c^
	T2	8.64 ± 0.12 ^b^	9.24 ± 0.12 ^a^	9.89 ± 0.19 ^c^	9.35 ± 0.34 ^bc^
	T3	9.05 ± 0.07 ^a^	9.31 ± 0.13 ^a^	10.47 ± 0.05 ^a^	9.81 ± 0.32 ^a^
	T4	8.99 ± 0.07 ^a^	9.15 ± 0.08 ^a^	10.15 ± 0.06 ^b^	9.76 ± 0.17 ^ab^
ΔE	C	-	1.11 ± 0.81 ^a^	1.57 ± 0.40 ^b^	1.61 ± 0.44 ^a^
	VC	-	0.95 ± 0.26 ^a^	2.16 ± 0.31 ^ab^	1.85 ± 0.26 ^a^
	T1	-	0.88 ± 0.22 ^a^	2.71 ± 0.47 ^a^	2.02 ± 0.02 ^a^
	T2	-	0.93 ± 0.32 ^a^	2.15 ± 0.34 ^ab^	1.53 ± 0.39 ^a^
	T3	-	0.55 ± 0.27 ^a^	2.24 ± 0.35 ^ab^	1.42 ± 0.45 ^a^
	T4	-	1.12 ± 0.36 ^a^	1.57 ± 0.21 ^b^	1.40 ± 0.88 ^a^

Note: Results were expressed as mean ± standard deviation, different superscript letters in each column represent significant differences (*p* < 0.05) in samples. Treatments: C: control, without antioxidant; VC: 0.025% sodium D-isoascorbate; T1 T2, T3, and T4: contained 0.06%, 0.12%, 0.18%, and 0.24% flavonoid extracts from *C. paliurus*, respectively.

**Table 4 foods-11-03721-t004:** Changes in pH of frankfurters during storage at 4 °C (n = 6).

Group	Day 1	Day 7	Day 14	Day 21
C	6.28 ± 0.03 ^c^	6.24 ± 0.01 ^d^	6.26 ± 0.02 ^c^	6.27 ± 0.01 ^c^
VC	6.37 ± 0.01 ^a^	6.34 ± 0.01 ^a^	6.35 ± 0.02 ^a^	6.38 ± 0.01 ^a^
T1	6.28 ± 0.02 ^c^	6.24 ± 0.01 ^d^	6.25 ± 0.01 ^c^	6.27 ± 0.00 ^c^
T2	6.33 ± 0.00 ^b^	6.30 ± 0.01 ^b^	6.30 ± 0.01 ^b^	6.34 ± 0.00 ^b^
T3	6.33 ± 0.01 ^b^	6.30 ± 0.01 ^b^	6.31 ± 0.01 ^b^	6.34 ± 0.00 ^b^
T4	6.27 ± 0.01 ^c^	6.27 ± 0.01 ^c^	6.24 ± 0.00 ^c^	6.34 ± 0.01 ^b^

Note: Results were expressed as mean ± standard deviation, different superscript letters in each column represent significant differences (*p* < 0.05) in samples. Treatments: C: control, without antioxidant; VC: 0.025% sodium D-isoascorbate; T1 T2, T3, and T4: contained 0.06%, 0.12%, 0.18%, and 0.24% flavonoid extracts from *C. paliurus*, respectively.

**Table 5 foods-11-03721-t005:** Changes of Texture parameters of frankfurters during storage at 4 °C (n = 6).

	Group	Day 1	Day 7	Day 14	Day 21
Firmness	C	3086.81 ± 752.42 ^a^	3171.03 ± 364.37 ^a^	3068.52 ± 210.17 ^a^	3615.85 ± 273.00 ^a^
(g)	VC	2733.77 ± 291.16 ^a^	2245.53 ± 759.71 ^b^	2188.06 ± 837.95 ^c^	3046.26 ± 133.80 ^bc^
	T1	2955.75 ± 396.19 ^a^	2847.49 ± 304.29 ^b^	2767.90 ± 176.49 ^ab^	3148.80 ± 306.33 ^b^
	T2	2724.59 ± 325.10 ^a^	2653.89 ± 317.15 ^b^	2953.38 ± 202.19 ^ab^	2753.21 ± 266.29 ^cd^
	T3	2666.20 ± 252.41 ^a^	2507.66 ± 250.34 ^ab^	2268.10 ± 233.90 ^c^	2662.89 ± 332.35 ^d^
	T4	2855.44 ± 416.03 ^a^	3112.30 ± 480.21 ^a^	2497.07 ± 260.61 ^bc^	3034.64 ± 307.55 ^bc^
Springiness	C	0.88 ± 0.01 ^a^	0.89 ± 0.03 ^a^	0.88 ± 0.01 ^a^	0.91 ± 0.01 ^a^
	VC	0.89 ± 0.03 ^a^	0.83 ± 0.22 ^a^	0.79 ± 0.26 ^a^	0.89 ± 0.03 ^ab^
	T1	0.86 ± 0.03 ^a^	0.90 ± 0.01 ^a^	0.89 ± 0.03 ^a^	0.92 ± 0.01 ^a^
	T2	0.86 ± 0.03 ^a^	0.87 ± 0.02 ^a^	0.88 ± 0.04 ^a^	0.88 ± 0.04 ^ab^
	T3	0.84 ± 0.05 ^a^	0.88 ± 0.01 ^a^	0.86 ± 0.05 ^a^	0.85 ± 0.05 ^b^
	T4	0.85 ± 0.05 ^a^	0.87 ± 0.03 ^a^	0.86 ± 0.04 ^a^	0.87 ± 0.02 ^ab^
Chewiness	C	2130.12 ± 499.71 ^a^	2232.20 ± 278.93 ^a^	2134.01 ± 154.73 ^a^	2602.14 ± 142.18 ^a^
	VC	1939.95 ± 195.54 ^a^	1637.58 ± 691.84 ^a^	1561.43 ± 714.85 ^c^	2186.00 ± 72.06 ^bc^
	T1	2031.69 ± 411.11 ^a^	2048.52 ± 230.96 ^abc^	1969.25 ± 118.72 ^abc^	2291.84 ± 234.07 ^b^
	T2	1832.71 ± 241.25 ^a^	1825.69 ± 209.69 ^abc^	2038.95 ± 214.90 ^ab^	1927.50 ± 245.28 ^cd^
	T3	1754.68 ± 245.52 ^a^	1723.23 ± 178.87 ^bc^	1573.09 ± 176.98 ^c^	1777.64 ± 330.80 ^d^
	T4	1903.15 ± 319.74 ^a^	2153.83 ± 398.91 ^ab^	1695.26 ± 238.97 ^bc^	2041.23 ± 240.69 ^bcd^

Note: Results were expressed as mean ± standard deviation, different superscript letters in each column represent significant differences (*p* < 0.05) in samples. Treatments: C: control, without antioxidant; VC: 0.025% sodium D-isoascorbate; T1 T2, T3, and T4: contained 0.06%, 0.12%, 0.18%, and 0.24% flavonoid extracts from *C. paliurus*, respectively.

**Table 6 foods-11-03721-t006:** Changes of total viable count (TVC) of frankfurters during storage (n = 6).

Group	Day 1	Day 7	Day 14	Day 21
C	0	2.87 ± 0.09 ^a^	3.21 ± 0.08 ^a^	4.19 ± 0.10 ^a^
Vc	0	0	0	2.71 ± 0.15 ^d^
T1	0	2.11 ± 0.26 ^d^	2.14 ± 0.15 ^d^	2.96 ± 0.06 ^c^
T2	0	2.39 ± 0.04 ^c^	2.67 ± 0.22 ^b^	2.98 ± 0.02 ^c^
T3	2.71 ± 0.12 ^a^	2.65 ± 0.14 ^ab^	2.86 ± 0.09 ^b^	3.14 ± 0.04 ^b^
T4	2.56 ± 0.09 ^b^	2.62 ± 0.06 ^bc^	2.80 ± 0.07 ^b^	2.88 ± 0.06 ^c^

Note: Results were expressed as mean ± standard deviation, different superscript letters in each column represent significant differences (*p* < 0.05) in samples. Treatments: C: control, without antioxidant; VC: 0.025% sodium D-isoascorbate; T1 T2, T3, and T4: contained 0.06%, 0.12%, 0.18%, and 0.24% flavonoid extracts from *C. paliurus*, respectively.

**Table 7 foods-11-03721-t007:** Effect of extract addition on sensory quality of frankfurters during storage at 4 °C (*n* = 10).

	Group	Day 1	Day 7	Day 14	Day 21
Color	C	7.3 ± 0.5 ^a^	7.4 ± 0.4 ^a^	7.5 ± 1.4 ^b^	7.1 ± 1.1 ^b^
	VC	7.3 ± 0.5 ^a^	7.4 ± 0.5 ^a^	8.1 ± 0.9 ^a^	7.8 ± 0.5 ^a^
	T1	7.1 ± 0.3 ^b^	7.5 ± 0.5 ^a^	8.1 ± 1.3 ^a^	7.6 ± 0.7 ^a^
	T2	7.2 ± 0.4 ^a^	7.5 ± 0.9 ^a^	8.3 ± 0.8 ^a^	7.5 ± 0.9 ^ab^
	T3	7.3 ± 0.9 ^a^	7.3 ± 0.7 ^a^	7.9 ± 1.3 ^ab^	7.0 ± 0.6 ^b^
	T4	7.3 ± 1.0 ^a^	7.4 ± 0.7 ^a^	7.6 ± 1.3 ^b^	7.0 ± 1.2 ^b^
Flavor	C	8.2 ± 0.4 ^a^	7.9 ± 0.7 ^a^	7.0 ± 1.2 ^b^	6.5 ± 1.5 ^c^
	VC	8.1 ± 0.6 ^ab^	8.2 ± 0.8 ^a^	7.4 ± 1.4 ^a^	7.3 ± 0.9 ^a^
	T1	8.0 ± 0.5 ^b^	7.7 ± 1.2 ^a^	7.4 ± 1.6 ^a^	7.1 ± 1.0 ^ab^
	T2	8.3 ± 0.5 ^a^	7.5 ± 1.2 ^b^	7.2 ± 1.6 ^ab^	7.2 ± 1.2 ^a^
	T3	7.4 ± 1.0 ^d^	7.7 ± 1.2 ^ab^	6.8 ± 0.9 ^b^	6.9 ± 1.2 ^b^
	T4	7.6 ± 0.5 ^c^	7.4 ± 1.2 ^b^	6.6 ± 1.3 ^c^	7.0 ± 1.2 ^b^
Juiciness	C	7.5 ± 0.5 ^a^	6.8 ± 1.0 ^b^	6.1 ± 1.1 ^b^	6.0 ± 1.3 ^b^
	VC	7.6 ± 0.5 ^a^	7.3 ± 1.2 ^a^	6.7 ± 1.1 ^a^	6.4 ± 1.8 ^a^
	T1	7.6 ± 0.5 ^a^	7.2 ± 1.2 ^a^	6.8 ± 1.2 ^a^	6.0 ± 1.2 ^a^
	T2	7.5 ± 0.5 ^a^	7.1 ± 1.1 ^a^	6.9 ± 0.8 ^a^	6.3 ± 1.9 ^a^
	T3	7.3 ± 0.5 ^a^	7.0 ± 1.4 ^ab^	6.2 ± 1.1 ^b^	6.3 ± 1.1 ^a^
	T4	7.4 ± 0.7 ^a^	6.7 ± 2.0b ^a^	6.5 ± 1.5 ^ab^	6.2 ± 1.4 ^ab^
Taste	C	7.5 ± 1.0 ^b^	7.7 ± 1.5 ^a^	7.0 ± 1.1 ^b^	6.7 ± 1.3 ^b^
	VC	7.9 ± 1.3 ^a^	7.8 ± 1.4 ^a^	7.6 ± 0.7 ^a^	7.4 ± 0.9 ^a^
	T1	7.4 ± 1.1 ^b^	7.4 ± 1.4 ^b^	7.1 ± 0.9 ^b^	7.0 ± 0.7 ^a^
	T2	7.5 ± 1.0 ^b^	7.7 ± 1.3 ^a^	7.4 ± 0.5 ^ab^	7.2 ± 1.2 ^a^
	T3	7.3 ± 0.9 ^b^	7.5 ± 1.2 ^b^	7.1 ± 0.9 ^b^	7.0 ± 1.1 ^a^
	T4	7.3 ± 0.7 ^b^	7.3 ± 1.5 ^b^	7.1 ± 1.0 ^b^	6.8 ± 1.2 ^ab^
Overall	C	8.1 ± 0.7 ^a^	7.8 ± 0.8 ^ab^	7.2 ± 1.0 ^b^	6.9 ± 1.1 ^b^
acceptability	VC	7.9 ± 0.6 ^a^	7.6 ± 0.5 ^b^	7.1 ± 0.5 ^b^	6.8 ± 1.0 ^b^
	T1	7.8 ± 0.8 ^a^	7.7 ± 0.9 ^b^	7.2 ± 0.9 ^b^	6.9 ± 0.7 ^b^
	T2	7.9 ± 0.7 ^a^	8.0 ± 0.7 ^a^	7.4 ± 0.5 ^a^	7.3 ± 1.2 ^a^
	T3	7.5 ± 0.5 ^a^	7.7 ± 1.1 ^b^	7.1 ± 1.0 ^b^	7.1 ± 0.9 ^b^
	T4	7.2 ± 0.4 ^a^	7.6 ± 1.3 ^b^	7.2 ± 0.8 ^b^	6.7 ± 1.0 ^b^

Note: Results were expressed as mean ± standard deviation, different superscript letters in each column represent significant differences (*p* < 0.05) in samples. Treatments: C: control, without antioxidant; VC: 0.025% sodium D-isoascorbate; T1 T2, T3, and T4: contained 0.06%, 0.12%, 0.18%, and 0.24% flavonoid extracts from *C. paliurus*, respectively.

## Data Availability

The data used to support the findings of this study can be made available by the corresponding author upon request.

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
