# Peer review of "Effects of Cyclocarya paliurus (Batal.) Extracts on Oxidative Stability and Sensory Quality in Meat Products (Frankfurters)"

_foods, 2022, doi:10.3390/foods11223721_

Round 1

Reviewer 1 Report

This manuscript presents an original study about the effects of Cyclocarya paliurus (Batal.) extracts on oxidative stability and sensory quality in meat products (Frankfurters).

The experimental data showed that different concentration of flavonoid extracts from C. paliurus affects the overall quality, fat oxidation, TVC and sensory quality of vacuum-packed cooked meat products (Frankfurters) over the period of 21 days of storage.

The scientific quality of the manuscript it rises to the scientific level of the Foods Journal. The technical quality of the manuscript is good in terms of how it was written and how the experimental results are presented. The style of expression reflects the scientific training of the authors. The manuscript is partly edited in accordance with the article drafting requirements.

The Abstract is concise and contains sufficient information to highlight the content of the article and the Introduction section provides a clear statement of the problem studied in the present manuscript.

The Materials and Methods section is satisfactorily presented and appropriate to the purpose of the research.

Results follow the guidelines described in the Author's Guide and are good discussed.

References are relevant and current and follow the journal’s format.

The conclusions of the article fully reflect the results of the given study.

Please find below some suggestions/comments that might help in improving the quality of the manuscript:

L 97. Please indicate the origin of all reagents, nutrient media and devices used in the research.

L 98. Please characterize the flavonoid extracts from C. paliurus that were used in the research from a sensory and physical point of view.

L 132. Why did the VC group contain 0.025% D-isoascorbate sodium?

L 133. Please clarify, the concentration of flavonoid extract from C. paliurus (0.06%, 0.12%, 0.18% and 0.24%) was reported to what?

L 177. Please indicate the name of the chromatic parameters: L*, a* and b*.

L 188. Sensory evaluation. Please present sensory characteristics for the evaluation of the finished product. The characteristics described in this section do not correspond to figure 2.

L 196. Statistical Analysis. Please indicate how many measurements you made for the statistical analysis.

L 222. In the note exclude - group C: control, without antioxidant.

L 225. Color analysis. Please carefully analyze whether the color of the added flavonoid extract from C. paliurus depending on its concentration can influence the color of the frankfurters.

L 274. pH analysis. How do you explain that the pH of the samples in groups 2 and 3 is higher than the control sample and in group 4 it is lower than the control sample.

L 310. Indicate the full name of the TPA abbreviation. Present the units of measurement of the texture parameters.

L 365. The figure 2 is not visible. I recommend that you present the evolution of the sensory characteristics and overall acceptability in a table.

Author Response

Point 1: L 97. Please indicate the origin of all reagents, nutrient media and devices used in the research.

Response 1: Thanks for your comments. We have added the information in our manuscript. Please kindly check (line 98-107, 134, 155, 177).

Point 2: L 98. Please characterize the flavonoid extracts from C. paliurus that were used in the research from a sensory and physical point of view.

Response 2: Thanks for your comments. We have added the details in our manuscript (line 98-99).

Point 3: L 132. Why did the VC group contain 0.025% D-isoascorbate sodium?

Response 3: Thanks for your comments. Our pre-experiment results showed that the antioxidant capacity of 0.025% sodium D-isoascorbate is equivalent to that of 0.02% BHT, so we chose 0.025% D-isoascorbate sodium as a positive control.

Point 4: L 133. Please clarify, the concentration of flavonoid extract from C. paliurus (0.06%, 0.12%,

0.18% and 0.24%) was reported to what?

Response 4: Thanks for your comments. Xie et al. found that the DPPH radical scavenging capacity of flavonoids from C. paliurus (concentration 0.1-0.8mg/mL) was always better than that BHT (line 71). The color and flavor of the extracts maybe affect the quality of the frankfurters, so the lower concentrations of 0.12%, 0.18% and 0.24% were selected. Meanwhile, the results of DPPH radical scavenging capacity showed that 0.06% flavonoid extracts from C. paliurus with effective DPPH scavenging activity, so we chose the concentration of 0.06% as a treatment group.

Point 5: L 177. Please indicate the name of the chromatic parameters: L*, a* and b*.

Response 5: Thanks for your comments. We have indicate the name of the chromatic parameters: L*, a* and b*. Please kindly check (line 251).

Point 6: L 188. Sensory evaluation. Please present sensory characteristics for the evaluation of the finished product. The characteristics described in this section do not correspond to figure 2.

Response 6: Thanks for your comments. We have added and changed the description of sensory evaluation. Please kindly check (line 204-217).

Point 7: L 196. Statistical Analysis. Please indicate how many measurements you made for the statistical analysis.

Response 7: Thanks for your comments. All measurements were performed in triplicate, and each treatment numbers were added in our manuscript. Please kindly check (line 125, 166, 174, 180, 193, 203, 244, 292. 319, 341, 369, 392. and 418).

Point 8: L 222. In the note exclude - group C: control, without antioxidant.

Response 8: Thanks for your comments. We have deleted. Please kindly check (line 246).

Point 9: L 225. Color analysis. Please carefully analyze whether the color of the added flavonoid extract from C. paliurus depending on its concentration can influence the color of the frankfurters.

Response 9: Thanks for your comments. On the 1st day, b * of frankfurters showed a trend of increasing with the concentration of flavonoid extractas, due to the influence of extracts color. The extract contains phenolic substances, the higher concentration of the extract, the higher composition of phenolic substances. Phenolic substances could be oxidized into quinones during storage, thus reducing the L* of frankfurters. We have added the analysis in our manuscript. Please kindly check (line 268-271, 280-282).

Point 10: L 274. pH analysis. How do you explain that the pH of the samples in groups 2 and 3 is higher than the control sample and in group 4 it is lower than the control sample.

Response 10: Thanks for your comments. We have added some analysis about this phenomenon in our manuscript. Please kindly check (line 301-305).

Point 11: L 310. Indicate the full name of the TPA abbreviation. Present the units of measurement of the texture parameters.

Response 11: Thanks for your comments. We have indicated the full name of the TPA abbreviation, and presented the units of measurement of the texture parameters in the table. Please kindly check (line 341).

Point 12: L 365. The figure 2 is not visible. I recommend that you present the evolution of the sensory characteristics and overall acceptability in a table.

Response 11: Thanks for your comments. We have presented the evolution of the sensory characteristics and overall acceptability in a table. Please kindly check (line 418).

Reviewer 2 Report

#1 The manuscript is missing the information about the chemical characterization of the end product (protein, fat, water....).

#2 The determination of texture was referred to the method of Zhou et al. althought this manuscript does not contain texture analysis at all ?!?! We can only guess that the researchers performed Texture Profile Analysis (TPA) testing.  Over the past few years TPA testing has been the cause of much concern. In general, TPA is a very popular method of testing, as it provides very quick calculation of parameters which are 'believed to correlate with sensory analysis'. The following is a set of points to consider when choosing TPA as your test procedure:

#a Size of Compression Probe versus Sample When the probe is larger than the sample, the forces registered are largely due to uniaxial compression. However, when the opposite is true, the forces derive largely from puncture, a combination of compression and shear. Various papers throughout the decades of using TPA have reported the use of probes both larger and smaller than the test samples. Early papers on TPA report the use of puncture probes, but in 1968 Prof. Malcolm Bourne was the first to adopt true uniaxial compression to perform TPA tests. Generally speaking, most recent work done on TPA uses compression probes of the same size as or larger than the sample size, so that the forces registered in such TPA tests are largely due to uniaxial compression forces and the whole of the sample piece is tested. Unfortunately, the authors opted for approximate dimension of the sample of 22mm in diameter  and we are not informed about the diameter of the probe.  Therefore, if the probe was smaller then the sample this shortcoming must be addressed and influence on the obtained results explained.

#b Extent of Deformation Another area of abuse is the degree of compression. The original TPA work used 80% strain. Most of Dr. Bourne's TPA research was conducted at 90% strain. The premise was that most foods should be chewed very fully in order to successively break up the mass until it is acceptable to swallow. If breaking up the food until it is palatable to swallow is the test objective, then by all means test products using strains approximating 66% to 80%. However, the authors decided to use 35% compression. They must explain the basis for such unusual decision?

#3 The need for standardized set of minimum reportable parameters for instrumental meat color evaluation still remains to be identified and incorporated in peer-reviewed journals guidelines for authors, as it was the case a decade ago. In the most recent review regarding meat color https://doi.org/10.1016/j.cofs.2021.02.012  the authors are proposing that all manuscripts containing instrumental color data must report on all instrumental details. Unfortunately, this manuscript missed to report about them.

#4 Lines 310-312: Delta E is a standard measurement — created by the Commission Internationale de l’Eclairage (International Commission on Illumination) — that quantifies the difference between two colors. The latest research indicates that a total color difference (ΔE) of approximately 1 is discriminable by consumers( https://doi.org/10.1016/j.meatsci.2022.108766 ) . The researchers need to calculate ΔE for their samples and discussion must include this new findings and conclusions should be revised accordingly.

#5 When setting a sensory panel, first step is its validation to confirm that the panel can work and be used in sensory studies. Also, validation of training (prior to sensory analysis), may result in exclusion of panelists due to discriminating problems. No information is provided regarding validation methods used and resulting activities applying to this sensory panel (https://doi.org/10.1111/jtxs.12616 ).

#6 Besides validating the panel (prior to sensory analysis), it is also important to assess panelists performance. Criteria for evaluating the attributes of a trained sensory panel and evaluation of the panel performance cover the following aspects: (i) is the panel capable of showing products differences / discriminate in-between samples; (ii) are the scores of panelists reliable (in-between replicates and over time in case of evaluating products over time); (iii) are results valid in terms of visible consensus between panelists and scoring in a similar way; and (iv) are they able to specify specific sensory attributes and sensations. Standard ISO 11132 outlines all four criteria as of equal importance: discriminability (linked with differences between products), homogeneity (consensus of the panel), repeatability (within sessions), and reproducibility (between session) as well as two-way ANOVA for panel performance (discrimination, homogeneity and repeatability) and one-way ANOVA for assessor performance (discrimination and repeatability). No information is provided regarding assessment of the panel performance used in this study.

Author Response

Point 1: The manuscript is missing the information about the chemical characterization of the end product (protein, fat, water....).

Response 1: Thanks for your comments, we have tested the changes of chemical characterization of frankfurters on the 1st, 7th, 14th, and 21st day of storage in various test groups, and found there was no significant difference during storage. So we don’t put the results in our manuscipt.

Point 2: The determination of texture was referred to the method of Zhou et al. althought this manuscript does not contain texture analysis at all ?!?! We can only guess that the researchers performed Texture Profile Analysis (TPA) testing.  Over the past few years TPA testing has been the cause of much concern. In general, TPA is a very popular method of testing, as it provides very quick calculation of parameters which are 'believed to correlate with sensory analysis'. The following is a set of points to consider when choosing TPA as your test procedure:

  1. Size of Compression Probe versus Sample When the probe is larger than the sample, the forces registered are largely due to uniaxial compression. However, when the opposite is true, the forces derive largely from puncture, a combination of compression and shear. Various papers throughout the decades of using TPA have reported the use of probes both larger and smaller than the test samples. Early papers on TPA report the use of puncture probes, but in 1968 Prof. Malcolm Bourne was the first to adopt true uniaxial compression to perform TPA tests. Generally speaking, most recent work done on TPA uses compression probes of the same size as or larger than the sample size, so that the forces registered in such TPA tests are largely due to uniaxial compression forces and the whole of the sample piece is tested. Unfortunately, the authors opted for approximate dimension of the sample of 22mm in diameter and we are not informed about the diameter of the probe.  Therefore, if the probe was smaller then the sample this shortcoming must be addressed and influence on the obtained results explained.
  2. Extent of Deformation Another area of abuse is the degree of compression. The original TPA work used 80% strain. Most of Dr. Bourne's TPA research was conducted at 90% strain. The premise was that most foods should be chewed very fully in order to successively break up the mass until it is acceptable to swallow. If breaking up the food until it is palatable to swallow is the test objective, then by all means test products using strains approximating 66% to 80%. However, the authors decided to use 35% compression. They must explain the basis for such unusual decision?

Response 2: Thanks for your careful review and comments. We have checked the reference of Zhou et al.,(Zhou, H.Y.; Deng S.L.; Zhou, C.Y.; Z, X.B.; Z, G.H. Effect of Fermented Blueberry Juice on Oxidative Stability and Quality Characteristics of Frankfurters. Food Sci (Chinese). 2019, 40, 69-76), which is published in food science (chinese), this paper contains texture analysis on page 71 section 1.3.8. Zhou et al. published another paper in Asian-Australas J Anim Sci (Effect of fermented blueberry on the oxidative stability and volatile molecule profiles of emulsion-type sausage during refrigerated storage), which does not contain texture analysis. please kindly check.

  1. The frankfurters were cut into 2cm high and 22mm diameter cylinders, and a probe of model P/50 was used for texture analysis. The diameter of P/50 is 50mm, which is larger than the sample. We have added the diameter of the probe in manuscipt (line 183), please kindly check.
  2. Our previous study shows that the frankfurter sample will be crushed, and its state will be damaged when the compression ratio exceeds 50%, so we chose 35% compression ratio to study whether the addition of antioxidants will affect the texture of frankfurters under the same pressure conditions.

Point3:  The need for standardized set of minimum reportable parameters for instrumental meat color evaluation still remains to be identified and incorporated in peer-reviewed journals guidelines for authors, as it was the case a decade ago. In the most recent review regarding meat color https://doi.org/10.1016/j.cofs.2021.02.012  the authors are proposing that all manuscripts containing instrumental color data must report on all instrumental details. Unfortunately, this manuscript missed to report about them.

Response3: Thanks for your careful review and comments, which is very helpful for us to improve the quality of this manuscipt. We have read the review paper and benefited a lot, which is a very excellent review paper, we have quoted it in our manuscript (line 250). The details of color instrument also added in our manuscipt (line 183). Please kindly check.

Point 4: Lines 310-312: Delta E is a standard measurement — created by the Commission Internationale de l’Eclairage (International Commission on Illumination) — that quantifies the difference between two colors. The latest research indicates that a total color difference (ΔE) of approximately 1 is discriminable by consumers( https://doi.org/10.1016/j.meatsci.2022.108766 ) . The researchers need to calculate ΔE for their samples and discussion must include this new findings and conclusions should be revised accordingly.

Response 4: Thanks for your comments. which is very helpful for us to improve the quality of this manuscipt. We have read the review paper and benefited a lot, which is a very excellent review paper, we have quoted it in our manuscript (line 189). We have calculated and analysised the changes of ΔE of frankfurters during storage in our manuscipt, please kindly check (line 191 and line 255-261).

Point5:  When setting a sensory panel, first step is its validation to confirm that the panel can work and be used in sensory studies. Also, validation of training (prior to sensory analysis), may result in exclusion of panelists due to discriminating problems. No information is provided regarding validation methods used and resulting activities applying to this sensory panel (https://doi.org/10.1111/jtxs.12616 ).

Response 5: Thanks for your comments. which is very helpful for us to improve the quality of this manuscipt. We have read the review paper and benefited a lot, which is a very excellent review paper to help reader to improve the understand of sensory analysis, we have quoted it in our manuscript (line 208). Meanwhile, we set out the criteria for selecting sensory evaluators in our manuscipt (line 206-213).

Point 6: Besides validating the panel (prior to sensory analysis), it is also important to assess panelists performance. Criteria for evaluating the attributes of a trained sensory panel and evaluation of the panel performance cover the following aspects: (i) is the panel capable of showing products differences / discriminate in-between samples; (ii) are the scores of panelists reliable (in-between replicates and over time in case of evaluating products over time); (iii) are results valid in terms of visible consensus between panelists and scoring in a similar way; and (iv) are they able to specify specific sensory attributes and sensations. Standard ISO 11132 outlines all four criteria as of equal importance: discriminability (linked with differences between products), homogeneity (consensus of the panel), repeatability (within sessions), and reproducibility (between session) as well as two-way ANOVA for panel performance (discrimination, homogeneity and repeatability) and one-way ANOVA for assessor performance (discrimination and repeatability). No information is provided regarding assessment of the panel performance used in this study.

Response 6: Thanks for your comments. which is very helpful for us to improve the quality of this manuscipt. Referring to Djekic et al., the members of sensory evaluation group had a great influence on the results, training and performance prescreening of the senaory panels are very important for foods sensory evaluation. we set out the criteria for selecting sensory evaluators in our manuscipt (line 206-213). In fact, panelists were trained in accordance with ISO 11132: 2021 prior to evaluating, in order to reduce the differences caused by individuals.. We have been added this information in our manuscript, please kindly check.

Round 2

Reviewer 2 Report

The authors have successfully addressed all the issues raised by reviewers. The manuscript can be published as is.